# Existence and Uniqueness of Solution to a Terminal Value Problem of First-Order Differential Equation

**Yuqiang Feng** [1,*]**, Qian Pan** [2] **and Jun Jiang** [2]

1    School of Science, Wuhan University of Science and Technology, Wuhan 430081, China
2    Hubei Province Key Laboratory of Systems Science in Metallurgical Process, Wuhan 430081, China
*    Correspondence: yqfeng6@126.com

**Abstract:** The terminal value problem of differential equations has an important application background. In this paper, we are concerned with the terminal value problem of a first-order differential equation. Some sufficient conditions are given to obtain the existence and uniqueness results of solutions to the problem. Firstly, some comparison lemmas are established; secondly, an iterative technique and fixed point method are used to set up the main results; Finally, an example is provided to illustrate the application of the main results.

**Keywords:** terminal value problem; existence; uniqueness; comparison lemma; solution

**MSC:** 34C99; 34A40; 34A45

## 1. Introduction

The terminal value problem (also called the final value problem, initial inverse problem, backward in time problem, abbrev. **TVP**) is an exciting topic within differential equations. It has important applications in many fields, such as aerospace science, mathematical economics, optimal control, and differential games, etc. For example, in aerospace science, the question of how to design the flight path of a spacecraft given its landing site on a planet can be reduced to the terminal value problem of a differential equation.

With the development of nonlinear functional analysis, scholars have made significant progress with the use of the fixed point theory method in the study of the terminal value problem of differential equations. For example, Wang [1] transformed the terminal value problem of fractional differential equations into initial value problems based on the shooting method, and then used the theoretical results of the initial value problem of fractional differential equations in solving the terminal value problem. Finally, the effectiveness of this method to solve the final value problem of fractional differential equations was verified by numerical simulation. Zhang [2] used Monch's fixed point theorem to study the terminal value problem of first-order differential equations in Banach space, obtained a new existence theorem under looser conditions, and improved and generalized some known results. Wang [3] studied the existence and uniqueness of the solution to the terminal value problem of first-order differential equations with discontinuous terms in Banach space by using semi-order theory and the mixed monotone iteration technique, without involving compact conditions, and presented an error estimate of the iterative sequence of approximations to the solution. Zhou [4] used the new comparison results and semi-order theory to study the existence of the minimum and maximum solutions of the terminal value problem of first-order nonlinear differential equations in Banach space, and improved and generalized some known results. In [5], combining the generalized quasi-linearization technique with the upper and lower solutions method, Yakar and Arslan obtained a unique solution to the fractional causal terminal value problem. In [6], Shah and Rehman established a sufficient condition for the existence and uniqueness of the solution of a class of fractional differential equations over infinite intervals. In [7], the authors

discussed the terminal value differential inequality, the existence of extreme value solutions of differential equations, and the corresponding comparison principle. In [8], Benchohra et al. presented the existence results and uniqueness of solutions for a class of boundary value problems of the terminal type for fractional differential equations with the Hilfer–Katugampola fractional derivative. The reasoning was mainly based upon different types of classical fixed point theorems, such as the Banach contraction principle and Krasnoselskii's fixed point theorem. In [9], Li et al. were concerned with the well-posedness and efficient numerical algorithm for a terminal value problem with a generalized Caputo fractional derivative. They investigated the existence and uniqueness of the solution of the terminal value problem and considered the continuous dependence of the solutions on the given data. In [10], Babak and Wu tempered fractional differential equations with terminal value problems. Discretized collocation methods on piecewise polynomial spaces were proposed for solving these equations. Regularity results were constructed on weighted spaces, and convergence order was studied.

The above results are mainly based on the properties of compact operators or increasing operators.

In this paper, we are concerned with the following **TVP**,

$$u'(t) = f(t, u(t)) \quad t \in [0, T], \quad u(T) = u_T,$$

where $T > 0, u_T \in R$ are two constants and $f : [0, T] \times R \to R$ is continuous.

By the properties of decreasing operators, we obtain the existence and uniqueness of the solution to this problem. Our contributions are the following:

(1) we present some comparison lemmas for (**TVP**);

(2) we establish the existence and uniqueness results of solutions for (**TVP**);

(3) we set up an iterative scheme of approximation solutions for (**TVP**).

The paper is organized as follows. In Section 2, some comparison lemmas are established; the existence and uniqueness results of (**TVP**) are presented in Section 3 via the iterative technique and fixed point method; an example shown in Section 4 illustrates the application of the results obtained.

## 2. Comparison Lemmas

The following comparison lemmas are of importance throughout this paper.

**Lemma 1.** *If $v \in C^1[0, T]$ satisfies*

$$v'(t) + \lambda v(t) \geqslant 0 \quad v(T) \leqslant 0 \quad t \in [0, T]$$

*where $\lambda \in R$ is a constant, then $v(t) \leqslant 0$ for $t \in [0, T]$.*

**Proof.** Since $v'(t) + \lambda v(t) \geqslant 0$, we have

$$e^{\lambda t} \big( v'(t) + \lambda v(t) \big) \geqslant 0$$

that is,

$$\big( v(t) e^{\lambda t} \big)' \geqslant 0$$

which implies that $v(t) e^{\lambda t}$ is increasing on $[0, T]$. Hence, for $\forall t \in [0, T]$,

$$v(t) e^{\lambda t} \leqslant v(T) e^{\lambda T} \leqslant 0$$

i.e., $v(t) \leqslant 0, t \in [0, T]$. $\quad \square$

**Lemma 2.** *Let $v, w \in C^1[0, T]$, and $\lambda \in R$ be a constant. If*

$$w'(t) + \lambda w(t) \leqslant v'(t) + \lambda v(t) \quad v(T) \leq u_T \leq w(T) \quad t \in [0, T],$$

*then $v(t) \leqslant w(t)$ for $t \in [0, T]$.*

**Proof.** Let $h(t) = v(t) - w(t)$, then we have

$$h'(t) + \lambda h(t) \geqslant 0 \quad h(T) \leqslant 0 \quad t \in [0, T].$$

By Lemma 1, we know $h(t) \leqslant 0, t \in [0, T]$, i.e., $v(t) \leqslant w(t)$ for $t \in [0, T]$.  □

**Lemma 3.** *Let $w \in C^1[0, T], h \in C[0, 1]$, and $\lambda \in R$ be a constant. If*

$$w'(t) + \lambda w(t) \leqslant h(t) \quad w(T) \geq u_T \quad t \in [0, T],$$

*then*

$$w(t) \geq u_T e^{\lambda(T-t)} - \int_t^T e^{\lambda(s-t)} h(s) \, ds$$

*for $t \in [0, T]$.*

**Proof.** If $v \in C^1[0, 1]$ is a solution to the following terminal value problem

$$v'(t) + \lambda v(t) = h(t) \quad v(T) = u_T \quad t \in [0, T],$$

then we have

1.
$$v(t) = u_T e^{\lambda(T-t)} - \int_t^T e^{\lambda(T-t)} h(s) \, ds$$

2.
$$w'(t) + \lambda w(t) \leq v'(t) + \lambda v(t), \quad v(T) = u_T \leq w(T) \quad t \in [0, T].$$

By Lemma 2, we obtain

$$w(t) \geq v(t) = u_T e^{\lambda(T-t)} - \int_t^T e^{\lambda(T-t)} h(s) \, ds.$$

□

## 3. Main Results

In this section, we give some sufficient conditions to ensure the existence and uniqueness of the (**TVP**).

Firstly, we transform the (**TVP**) to a fixed point problem; secondly, we construct an iterative sequence by the integral operator; finally, by using comparison lemmas, we verify that the sequence is uniformly convergent to the unique solution of the (**TVP**).

Let $u, v \in C[0, T]$; if $u(t) \leq v(t)$ for $\forall t \in [0, T]$, we denote $u \leq v$. The order interval $[u, v] = \{x \in C[0, T] | u(t) \leq x(t) \leq v(t), \forall t \in [0, T]\}$.

The main result of this paper is the following.

**Theorem 1.** *Let us say that there exist $v, w \in C^1[0, T], v \leqslant w$ and a constant $\lambda$ such that*
1. *for $\forall t \in [0, T], x, y \in [v, w], x \leqslant y$,*

$$f(t, y(t)) - f(t, x(t)) \geqslant -\lambda(y(t) - x(t))$$

2. *for $\forall t \in [0, T], 0 \leqslant \ell \leqslant 1$ and $x, y \in [v, w]$,*

$$f(t, lx(t) + (1-l)y(t)) \geqslant lf(t, x(t)) + (1-l)f(t, y(t))$$

3.  *for* $\forall t \in [0, T]$,
$$(v + w)'(t) + \lambda(v - w)(t) \geqslant 2f(t, w(t))$$
$$f(t, v(t)) \geqslant w'(t) + \lambda(w - v)(t) \tag{1}$$

4.  $v(T) = u_T = w(T)$.

*Then, (TVP) has a unique solution $\tilde{x}$ satisfying $v(t) \leq \tilde{x}(t) \leq w(t), t \in [0, T]$ (abbr. $\tilde{x} \in [v, w]$).*

**Proof.** Let $x \in C[0, 1]$. If $h \in C^1[0, T]$ be a solution to the following terminal value problem:
$$h'(t) + \lambda h(t) = f(t, x(t)) + \lambda x(t), \ h(t) = u_T$$

Then,
$$h(t) = u_T e^{\lambda(T-t)} - \int_t^T e^{\lambda(T-t)} [f(s, x(s)) + \lambda x(s)] \, ds$$

Define a mapping $T$ on $C[0, T]$ as follows:
$$(Tx)(t) = u_T e^{\lambda(T-t)} - \int_t^T e^{\lambda(s-t)} [f(s, x(s)) + \lambda x(s)] \, ds, \ \ x \in C[0, 1].$$

It is easy to verify that $T$ maps $C[0, T]$ into $C[0, T]$, and (TVP) has a solution if and only if $T$ has a fixed point in $C[0, T]$.

By Assumptions (1) and (2), we know that $T$ is decreasing and convex on $[v, w]$.

By the first inequality in (1), we have
$$\begin{cases} \left(\dfrac{v+w}{2}\right)'(t) + \lambda\left(\dfrac{v+w}{2}\right) \geqslant f(t \cdot w(t)) + \lambda w(t), \\ \left(\dfrac{v+w}{2}\right)(T) = u_T. \end{cases}$$

Due to the second inequality in (1), we obtain
$$\begin{cases} w'(t) + \lambda w(t) \leqslant f(t, v(t)) + \lambda v(t), \\ w(T) = u_T. \end{cases}$$

Let $x_0(t)$ be a solution to the following terminal value problem:
$$\begin{cases} u'(t) + \lambda u(t) = f(t, w(t)) + \lambda w(t), \\ u(T) = u_T, \end{cases}$$

Then,
$$x_0(t) = u_T e^{\lambda(T-t)} - \int_t^T e^{\lambda(s-t)} [f(s, w(s)) + \lambda w(s)] \, ds.$$

Construct an iterative sequence $\{x_n(t)\}$ as follows:
$$\begin{cases} x'_{n+1}(t) + \lambda x_{n+1}(t) = f(t, x_n(t)) + \lambda x_n(t) \\ x_{n+1}(T) = u_T \end{cases} \quad n = 0, 1, 2, \cdots$$

i.e.,
$$x_{n+1}(t) = u_T e^{\lambda(T-t)} - \int_t^T e^{\lambda(s-t)} [f(s, x_n(s)) + \lambda x_n(s)] \, ds.$$

In what follows, we prove that $\{x_n\}$ is a Cauchy sequence in $C[0, T]$, and converges to the solution of the (**TVP**) in $C[0, T]$.

**Step 1.** We assert

$$w(t) \geqslant x_0(t) \geqslant \left(\frac{w+v}{2}\right)(t) \geqslant v(t) \quad t \in [0, T].$$

In virtue of

$$\left(\frac{w+v}{2}\right)'(t) + \lambda\left(\frac{v+w}{2}\right)(t) \geqslant x_0'(t) + \lambda x_0(t)$$

and

$$\left(\frac{u+v}{2}\right)(T) = u_T = x_0(T)$$

and Lemma 2, we obtain

$$x_0(t) \geqslant \left(\frac{w+v}{2}\right)(t).$$

Moreover,

$$\begin{aligned}
w'(t) + \lambda w(t) \leqslant & f(t, v(t)) + \lambda v(t) \\
\leqslant & f(t, w(t)) + \lambda w(t) \\
= & x_0'(t) + \lambda x_0(t), \\
w(T) = & u_T = x_0(T).
\end{aligned}$$

Hence, by Lemma 2, we have $x_0(t) \leqslant w(t)$, and

$$w \geqslant x_0 \geqslant \frac{w+v}{2} \geqslant v.$$

**Step 2.** For $n = 0, 1, 2, \cdots$,

$$w(t) \geqslant x_{2n+1}(t) \geqslant x_{2n}(t) \geqslant \left(\frac{w+v}{2}\right)(t) \geqslant v(t).$$

On the one hand, since

$$\begin{aligned}
x_1(t) = & u_T e^{\lambda(T-t)} - \int_t^T e^{\lambda(s-t)}[f(s, x_0(s)) + \lambda x_0(s)]\, ds \\
= & (Tx_0)(t)
\end{aligned}$$

and $T$ is decreasing, we obtain

$$Tw \leqslant Tx_0 = x_1 \leqslant T\left(\frac{\omega+v}{2}\right) \leqslant Tv.$$

Noting that $x_0 = Tw$, we have

$$x_0 \leqslant x_1.$$

On the other hand, by the second inequality in (1), we have

$$w'(t) + \lambda w(t) \leqslant f(t, v(t)) + \lambda v(t) \leqslant x_1'(t) + \lambda x_1 t = f(t_1, x_0(t)) + \lambda x_0(t)$$

which means

$$x_1'(t) + \lambda x_1(t) \geqslant w'(t) + \lambda w'(t)$$

By the comparison Lemma 2, there holds $x_1(t) \leqslant w(t)$. Hence,

$$x_0 \leqslant x_1 \leqslant w.$$

Noting that $x_0 \geqslant \frac{w+v}{2}$, we have $w \geqslant x_1 \geqslant x_0 \geqslant \frac{w+v}{2} \geqslant v$, which implies that the assertion holds for $n = 0$.

Suppose that when $n = k$,

$$w(t) \geqslant x_{2k+1}(t) \geqslant x_{2k}(t) \geqslant \left(\frac{v+w}{2}\right)(t) \geqslant v(t)$$

then

$$
\begin{aligned}
f(t, w(t)) + \lambda w(t) &\geqslant f(t, x_{2k+1}(t)) + \lambda x_{2k+1}(t) \\
&\geqslant f(t, x_{2k}(t)) + \lambda x_{2k}(t) \\
&\geqslant f(t \cdot v(t)) + \lambda v(t).
\end{aligned}
$$

Hence, we have

$$
\begin{aligned}
\left(\frac{v+w}{2}\right)'(t) + \lambda \left(\frac{v+w}{2}\right)(t) &\geqslant x'_{2k+2}(t) + \lambda x_{2k+2}(t) \\
&\geqslant x'_{2k+1}(t) + \lambda x_{2k+1}(t) \\
&\geqslant w'(t) + \lambda w(t).
\end{aligned}
$$

By the comparison Lemma 2,

$$\frac{v+w}{2} \leqslant x_{2k+2}(t) \leqslant x_{2k+1}(t) \leqslant w(t).$$

Repeating this process, we can verify

$$\frac{v+w}{2} \leqslant x_{2k+2}(t) \leqslant x_{2_{k+3}}(t) \leqslant w(t),$$

which means the assertion holds for $n = k + 1$.

Hence, for all $n$, there holds

$$w(t) \geqslant x_{2n+1}(t) \geqslant x_{2n}(t) \geqslant \left(\frac{v+w}{2}\right)(t) \geqslant v(t)$$

**Step 3.** $\{x_{2n}(t)\}$ is increasing, while $\{x_{2n+1}(t)\}$ is decreasing.

Since

$$u(t) \leqslant \left(\frac{v+w}{2}\right)(t) \leqslant x_1(t) \leqslant w(t),$$

then

$$
\begin{aligned}
f(t, v(t)) + \lambda v(t) &\leqslant f(t, x_1(t)) + \lambda x_1(t) \\
&\leqslant f(t, \lambda w(t)) + \lambda w(t))
\end{aligned}
$$

and

$$w'(t) + \lambda w(t) \leqslant x'_2(t) + \lambda x_2(t) \leqslant x'_0(t) + \lambda x_0(t)$$

By the comparison Lemma 2,

$$x_0(t) \leqslant x_2(t) \leqslant w(t)$$

In a similar way to Step 2, we can prove

$$\{x_{2n}(t)\} \text{ is increasing}, \{x_{2n+1}(t)\} \text{ is decreasing}.$$

**Step 4.** $\{x_n(t)\}$ is uniformly convergent on $[0, 1]$.

By Steps 1–3, we know that $\{x_n(t)\}$ satisfies

$$v(t) \leqslant \left(\frac{v+w}{2}\right)(t) \leqslant x_0(t) \leqslant x_2(t) \leqslant \cdots \leqslant x_{2n}(t) \leqslant \cdots \leqslant x_{2n+1}(t) \leqslant \cdots \leqslant x_1(t) \leqslant w(t)$$

Let $Z_n(t) = x_n(t) - v(t)$. We have

$$0 \leqslant \left(\frac{w-v}{2}\right)(t) \leqslant Z_0(t) \leqslant Z_2 t \leqslant \cdots \leqslant Z_{2n}(t) \leqslant \cdots \leqslant Z_{2n+1}(t) \leqslant \cdots \leqslant Z_1(t)$$
$$\leqslant (w-v)(t).$$

Define

$$r_n = \sup\{r \in R \mid Z_{2n}(t) \geqslant r Z_{2n+1}(t)\}$$

Then, the sequence $\{r_n\}$ is well defined, $\frac{1}{2} \leqslant r_n \leqslant 1$, and $\{r_n\}$ is increasing.

Since

1.

$$Z_{2n}(t) \geqslant \frac{1}{2}(w-v)(t) \geqslant \frac{1}{2}Z_{2n+1}(t)$$

we have $r_n \geqslant \frac{1}{2}$

2.

$$Z_{2n}(t) \geqslant Z_{2n+1}(t)$$

and we obtain $r_n \leqslant 1$.

3. If $r$ satisfies $Z_{2n}(t) \geqslant r Z_{2n+1}(t)$, then the monotonicity of $\{Z_{2n}\}$ and $\{Z_{2n+1}\}$ implies

$$Z_{2n+2}(t) \geqslant Z_{2n}(t) \geqslant r Z_{2n+1}(t) \geqslant r Z_{2n+3}(t)$$

i.e., $\{r \in R \mid Z_{2n}(t) \geqslant r Z_{2n+1}(t)\} \subset \{r \in R \mid Z_{2n+2}(t) \geqslant r Z_{2n+3}(t)\}$.

By (1–3), we know that $\{r_n\}$ is convergent. Denote $r_0 = \lim\limits_{n \to \infty} r_n$.

By the comparison Lemma 3,

$$
\begin{aligned}
Z_{2n+3}(t) &\leqslant Z_{2n+1}(t) = x_{2n+1}(t) - v(t) \\
&= u_T e^{\lambda(T-t)} - \int_t^T e^{\lambda(s-t)}[f(s, x_{2n}(s)) + \lambda x_{2n}(s)]\, ds - v(t) \\
&= u_T e^{\lambda(T-t)} - \int_t^T e^{\lambda(s-t)}[f(s, Z_{2n}(s) + v(s))) + \lambda(Z_{2n} + v(s))\, ds - v(t) \\
&\leqslant u_T e^{\lambda(T-t)} - \int_t^T e^{\lambda(s-t)}[f(s, (r_n Z_{2n+1} + v)(s)) + \lambda(r_n Z_{2n+1} + v)(s)]\, ds - v(t) \\
&= u_T e^{\lambda(T-t)} - \int_t^T e^{\lambda(s-t)}\begin{bmatrix} f(s, (r_n x_{2n+1} + (1-r_n)v(s))) + \\ \lambda(r_n x_{2n+1} + (1-r_n)v(s)) \end{bmatrix} ds - v(t) \\
&\leqslant u_T e^{\lambda(T-t)} - \int_t^T e^{\lambda(s-t)}\begin{bmatrix} r_n(f(s, x_{2n+1}(s)) + \lambda x_{2n+1}(s)) + \\ (1-r_n)(f(s, v(s)) + \lambda v(s)) \end{bmatrix} ds - v(t) \\
&= r_n\left[ u_T e^{\lambda(T-t)} - \int_t^T e^{\lambda(s-t)}[f(s, x_{2n+1}(s)) + \lambda x_{2n+1}(s)\, ds - v(t)] \right] \\
&\quad + (1-r_n)\left[ u_T e^{\lambda(T-t)} - \int_t^T e^{\lambda(s-t)}[f(s, v(s)) + \lambda v(s)]\, ds - v(t) \right] \\
&\leqslant r_n[x_{2n+2}(t) - v(t)] + (1-r_n)[w(t) - v(t)] \\
&= r_n Z_{2n+2} + 2(1-r_n)\left(\frac{w-v}{2}\right)(t) \\
&\leqslant r_n Z_{2n+2} + 2(1-r_n)Z_{2n+2} \\
&= (2 - r_n)Z_{2n+2},
\end{aligned}
$$

then

$$r_{n+1} = \sup\{r \in R \mid Z_{2n+3}(t) \geqslant r Z_{2n+2}(t)\} \geqslant \frac{1}{2 - r_n}.$$

Taking the limit on both sides, we obtain

$$r_0 \geqslant \frac{1}{2 - r_0}.$$

Noting that $\frac{1}{2} \leqslant r_0 \leqslant 1$, we know $r_0 = 1$.

Then, for an even number $p$,

$$0 \leqslant Z_{2n+p} - Z_{2n} \leqslant Z_{2n+1} - Z_{2n} \leqslant (1 - r_n)Z_{2n+1} \leqslant (1 - r_n)(v - u).$$

Since $r_n \to 1$, $\{Z_{2n}\}$ is convergent. In a similar way, we obtain that $\{Z_{2n+1}\}$ is convergent, and

$$\lim_{n \to \infty} Z_{2n} = \lim_{n \to \infty} Z_{2n+1}.$$

Hence, $\{Z_n\}$ is convergent.

Let $Z = \lim\limits_{n \to \infty} Z_n$ and $\bar{x} = Z + v$, and then

$$
\begin{aligned}
\lim_{n \to \infty} x_n(t) &= \bar{x}(t) \\
&= \lim_{n \to \infty} x_{n+1}(t) \\
&= \lim_{n \to \infty} \left[ u_T e^{\lambda(T-t)} - \int_t^T [f(s, x_n(s)) + \lambda x_n(s)] \, ds \right] \\
&= u_T e^{\lambda(T-t)} - \int_t^T [f(s, \bar{x}(s)) + \lambda \bar{x}(s)] \, ds \\
&= (T\bar{x})(t),
\end{aligned}
$$

which means that $\bar{x}(t)$ is a fixed point of $T$.

**Step 5.** $\bar{x}$ is the unique fixed point of $T$ in $[v, w]$.

In fact, if $\tilde{x}$ is a fixed point of $T$ in $[v, w]$, then

$$
\begin{aligned}
v \leqslant \tilde{x} \leqslant w \Rightarrow & Tv \geqslant T\tilde{x} \geqslant Tw \\
\Rightarrow & w \geqslant Tv \geqslant \tilde{x} \geqslant x_0.
\end{aligned}
$$

Continuing this process, we have

$$x_{2n}(t) \leqslant \tilde{x}(t) \leqslant x_{2n+1}(t).$$

Taking the limit on both sides, we obtain

$$\tilde{x}(t) = \bar{x}(t).$$

Hence, $\bar{x}$ is the unique fixed point of $T$ in $[v, w]$, i.e., $\bar{x}$ is the unique solution of (TVP) in $[v, w]$. $\square$

**Remark 1.** *Let*

$$y_0(t) = u_T e^{\lambda(T-t)} - \int_t^T e^{\lambda(s-t)} [f(s, v(s)) + \lambda v(s)] \, ds$$

*Define*

$$y_{n+1}(t) = u_T e^{\lambda(T-t)} - \int_t^T e^{\lambda(s-t)} [f(s, y_n(s)) + \lambda y_n(s)] ds, n = 0, 1, 2, \cdots$$

*In the same way as in Theorem 1, we can prove that $\{y_n\}$ is uniformly convergent to $\bar{x}$ on $[0, 1]$.*

**Corollary 1.** *Assume that there exist two constants $c > 0$ and $\lambda \in R$ satisfying the following:*

1. $f(t,0) - \lambda c(T-t) \geqslant -c \geqslant 2f(t,c(T-t)) + \lambda c(T-t);$
2. *for $\forall t \in [0,T]$, $f(t,\cdot)$ is concave;*
3. *for $\forall t \in [0,T]$, $x,y \in [0,c(T-t)]$, $x \leqslant y$,*

$$f(t,y) - f(t,x) \geqslant -\lambda(y-x),$$

*and then (TVP)*

$$\begin{cases} x'(t) = f(t,x(t)) \\ x(T) = 0 \end{cases}$$

*has a unique solution $\bar{x}(t)$ satisfying*

$$0 \leqslant \bar{x}(t) \leqslant c(T-t),\ t \in [0,T].$$

**Proof.** Choose $v(t) = 0$, $w(t) = c(T-t)$, and we can verify that all conditions of Theorem 1 are fulfilled. $\square$

Consider the following terminal value problem.

$$\begin{cases} x'(t) = t + g(x(t))\ t \in [0,1] \\ x(1) = 0 \end{cases}$$

**Corollary 2.** *Let $g \in C^2[0,1]$. If the following conditions are satisfied*

1. $g(0) \geq -2;$
2. *for $\forall x \in [0,1]$, $g(x) \leq \frac{3}{2}x - \frac{3}{2};$*
3. *for $\forall x \in [0,1]$, $g'(x) \geq 1;$*
4. *for $\forall x \in [0,1]$, $g''(x) \leq 0$.*

*Then the above (TVP) has a unique solution $\bar{x}(t)$ satisfying*

$$0 \leqslant \bar{x}(t) \leqslant 1-t, t \in [0,1].$$

**Proof.** Let

$$f(t,x) = t + g(x).$$

and $c = T = 1, \lambda = -1$; we can verify that

$$\begin{aligned} f(t,0) - \lambda c(T-t) &\geq t + (-2) + 1 - t \\ &= -1 = -c \\ 2f(t,c(T-t)) + \lambda c(T-t) &= 2[t + g(1-t)] - (1-t) \\ &\leq 2[t + \frac{3}{2}(1-t) - \frac{3}{2}] - (1-t) \leqslant -1 = -c, \end{aligned}$$

which implies that assumption (1) of Corollary 1 is satisfied.

Moreover,

$$f''_{xx}(t,x) = g''(x) \leq 0$$

means that $f(t,.)$ is concave, i.e., Assumption (2) of Corollary 1 is fulfilled. Noting that

$$f'_x(t,x) = g'(x) \geqslant 1, \quad x \in [0,1],$$

hence, $f$ meets condition (3) of Corollary 1.

By Corollary 1, we know that this terminal problem has a unique solution $\bar{x}(t)$ satisfying

$$0 \leqslant \bar{x}(t) \leqslant 1 - t.$$

□

## 4. Application

**Example 1.** *Let $g(x) = x + sinx - 2$. We can verify that all assumptions of Corollary 2 hold. Hence, the terminal value problem*

$$\begin{cases} x'(t) = t + x(t) + sinx(t) - 2 \\ x(1) = 0 \end{cases}$$

*has a unique solution $\bar{x}(t)$ satisfying*

$$0 \leqslant \bar{x}(t) \leqslant 1 - t.$$

Let $T = 1, \lambda = -1, v_0(t) = 0$. Define

$$v_{n+1}(t) = u_T e^{\lambda(T-t)} - \int_t^T e^{\lambda(s-t)}[f(s, v_n(s)) + \lambda v_n(s)]ds, n = 0, 1, 2, \cdots .$$

Then, the approximate solutions of the above TVP are

$$v_1(t) = (t - 2)\left[e^{(t-1)} - 1\right]$$
$$v_2(t) = \left\{t + \sin\left\{(t - 2)\left[e^{(t-1)} - 1\right]\right\} - 2\right\} \cdot \left[e^{(t-1)} - 1\right]$$
$$\cdots$$

The image of the approximate solutions of $v_1, v_2$ is the Figure 1.

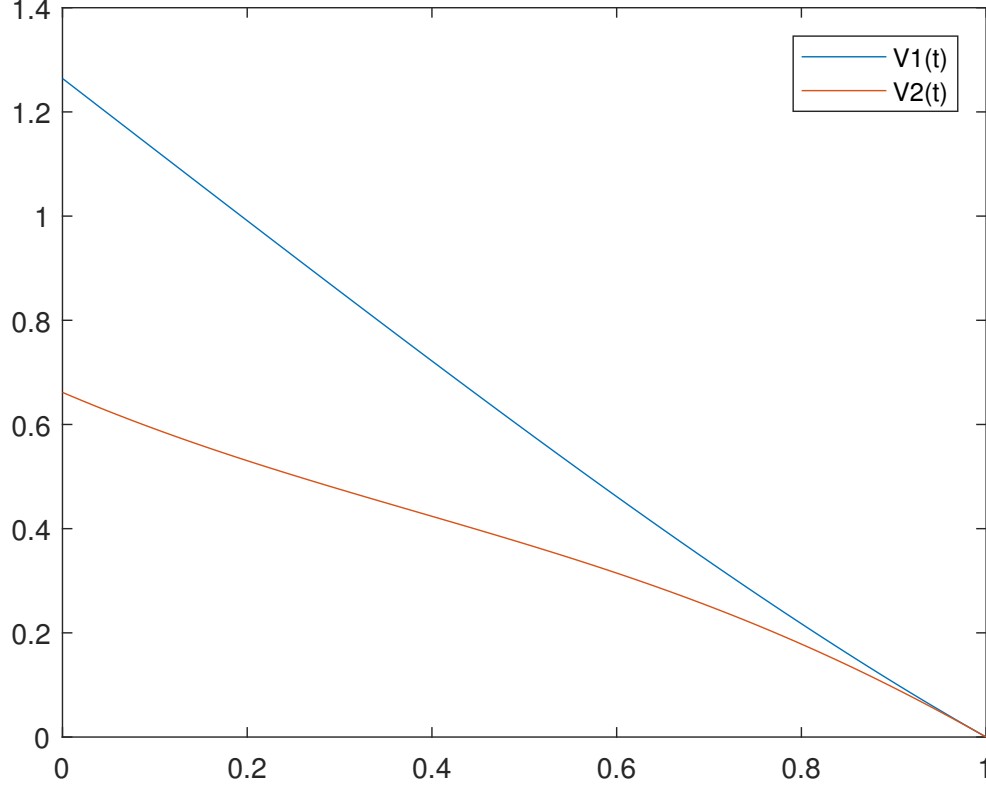

**Figure 1.** Image of the approximate solutions of $v_1, v_2$.

## 5. Discussion

In this paper, we have constructed an iterative monotone sequence and verified that this sequence is convergent to a solution of problem (**TVP**). Other assumptions ensure the uniqueness of this solution. Our uniqueness result is a **local result**, which means that the problem may have multiple solutions in the space $C[0, T]$.

## 6. Conclusions

In this paper, we have used comparison lemmas, an iterative technique and a fixed point method to obtain the existence and uniqueness results of solutions for a terminal value problem of the first-order differential equation. Our discussion lies in a bounded interval. It is an interesting problem to extend the study to an unbounded interval, i.e.,

$$\begin{cases} x'(t) & = f(t, x(t)) \\ x(\infty) & = \lim_{t \to \infty} x(t) = u_\infty \end{cases}$$

We will try to find appropriate conditions to ensure the existence and uniqueness of the solution to the above problem.

**Author Contributions:** Methodology, Y.F. and Q.P.; writing—original draft preparation, Q.P.; writing—review and editing, Y.F. and J.J.; funding acquisition, Y.F. All authors have read and agreed to the published version of the manuscript.

**Funding:** This research is partially supported by the National Natural Science Foundation of China (72031009).

**Institutional Review Board Statement:** Not applicable.

**Informed Consent Statement:** Not applicable.

**Data Availability Statement:** Not applicable.

**Acknowledgments:** The authors thank the anonymous referees for their valuable constructive comments and suggestions, which improved the quality of this paper in the present form.

**Conflicts of Interest:** The authors declare no conflict of interest.

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
