# Peer review of "Existence and Uniqueness of Solution to a Terminal Value Problem of First-Order Differential Equation"

_axioms, doi:10.3390/axioms11090435_

Round 1

Reviewer 1 Report

See attached file.

Author Response

Reply to Reviewer 1’s Comments

Thank you for your helpful suggestions.

According to your comments, we have made the following corrections.

  1. Please highlight your contribution separately.

Yes, we do. We added point-by-point the main contributions at the end of the Introduction section.

  1. Please indicate the difficulties that have been overcome in solving the

The previous research based on the properties of compact operators or increasing operators. The integral operator defined in this paper is a decreasing operator.

  1. Please indicate the relevance of the problem by citing literature from
    recent years.

Yes, we do. We added three papers in recent years. They focused on TVP of fractional differential equations and concerned with numerical solutions of the problems.

  1. Please add a conclusion.

  Yes, we do. We added the conclusion at the last section.

Reviewer 2 Report

Paper deals with the important question of terminal value problem. The authors considered the idea of the first-order differential equation. 

But from my side, I see a few points, which could be updated:

1. Abstract should be extended by the obtained results of the paper.

2.  It would be good to add point-by-point the main contributions at the end of the Introduction section.
3. I would suggest working on methods sections to make it better structured, provide comparisons and visualize steps, in a clearer way.

4. The results section should be extended using: 1) numerical results obtained in the paper; 2) limitations of the proposed approach; 3) prospects for future research.

4. The paper hasn’t any Discussions.
5. The paper hasn’t any Conclusions.

6. A lot of references are outdated and unlinked. Please fix it by using 3-5 years old papers in high-impact journals.

7. Paper does not meet guideline format.

Author Response

Reply to Reviewer 2’s Comments

Thank you for your helpful suggestions.

According to your comments, we have made the following corrections.

  1. Abstract should be extended by the obtained results of the paper.

Yes, we do.

  1. It would be good to add point-by-point the main contributions at the end of the Introduction section.

Yes, we do.

  1. I would suggest working on methods sections to make it better structured, provide comparisons and visualize steps, in a clearer way.

Yes, we do. We try to make the proof steps in a clearer way.

  1. The results section should be extended using: 1) numerical results obtained in the paper; 2) limitations of the proposed approach; 3) prospects for future research.

Thank you for your help. We have done the following

  • We presented the iterative scheme of solutions, computed  the approximation solutions and draw the figure of the approximation solutions of  a TVP,  see in Section 4;
  • We discussed the limitation of the proposed approach in Section 5;
  • We introduced the future research in Section 6.
  1. The paper hasn’t any Discussions.

We added the Discussion section.

  1. The paper hasn’t any Conclusions.

We added the Conclusion section.

  1. A lot of references are outdated and unlinked. Please fix it by using 3-5 years old papers in high-impact journals.

Yes, we do. We added three papers in recent three years.

  1. Paper does not meet guideline format.

We revised the format.

Reviewer 3 Report

Report on manuscript

Existence and uniqueness of solution to a terminal value problem of first-order differential equation

by

Yuqiang Feng, Qian Pan, Jun Jiang

·       Overview of manuscript

The authors presented the application of comparison lemmas, iterative technique, and fixed-point method, to obtain the existence and uniqueness results of solutions for a terminal value problem of first-order differential equation.

·       Comments on text

1.     New contribution

The authors need to talk about all the new contributions of their study.

·       English

The English in this paper is ok but we can see some grammatical issues such as “an terminal value” and so on.

Comments

(a)    The authors require to improve the literature review.

(b)   This manuscript does not include any discussion or conclusion part. Please provide these sections.

(c)    The authors may need to explain more about terminal problem in application.

(d)   That would be better if authors can briefly imply to the applications of comparison lemmas, iterative technique and fixed point method.

(e)    Please explain the contribution or novelty of this work.

(f)    Any numerical simulation to validate the results?

·       Recommendation

After all the revisions, I can recommend acceptance of this paper.

Author Response

Reply to Reviewer 3’s Comments

Thank you for your helpful suggestions.

According to your comments, we have made the following corrections.

  1. New contribution

The authors need to talk about all the new contributions of their study.

We added point-by-point the main contributions at the end of the Introduction section.

  1.      English

The English in this paper is ok but we can see some grammatical issues such as “an terminal value” and so on.

We have checked the full text and revised the grammar and vocabulary.

Comments

  • The authors require to improve the literature review.

We tried to improve the literature review.

  • This manuscript does not include any discussion or conclusion part. Please provide these sections.

We added the discussion and conclusion sections.

  • The authors may need to explain more about terminal problem in application.

We presented a brief explanation of TVP at the beginning of Introduction section.  

  • That would be better if authors can briefly imply to the applications of comparison lemmas, iterative technique and fixed point method.

We explained the applications of comparison lemmas, iterative technique and fixed point method at the beginning of Section 3.

  • Please explain the contribution or novelty of this work.

We added point-by-point the main contributions at the end of the Introduction section.

  • Any numerical simulation to validate the results?

In Section 4, we presented the iterative scheme of solutions, computed the approximation solutions and draw the figure of the approximation solutions of a TVP.

Round 2

Reviewer 2 Report

The authors took into account all my comments and accordingly improved the text of the article, which is now of better quality and significantly improved. My recommendation is to accept this paper for publishing as it is.

Reviewer 3 Report

The authors could address the comments.